# Masked face matching benefits from isolated facial features

**Mengying Zhang**\*, **Melanie Sauerland**, **Anna Sagana**

Section Forensic Psychology, Department of Clinical Psychological Science, Faculty of Psychology and Neuroscience, Maastricht University, Maastricht, Limburg, Netherlands

\* mengying.zhang@maastrichtuniversity.nl

## Abstract

Verifying the identity of an unfamiliar person is a difficult task, especially when targets wear masks that cover most of their faces. This presents a major challenge for law enforcement in border control, security, and criminal investigations. Therefore, we aim to explore ways to improve face-matching performance when a face is heavily masked. In two experiments, we investigated whether face-matching performance can benefit from the presentation of isolated facial features, namely the eyes (Experiment 1) and the mouth (Experiment 2), when a target face is masked. Participants viewed pairs of faces and determined whether they belonged to the same person or different people. In congruent pairs, participants matched a full-face image to another full-face image or a masked image to an isolated facial feature. In incongruent pairs, participants matched a full-face image to an image of the eyes or the mouth only or to a masked image. Matching accuracy was significantly better in congruent than incongruent pairs. Interestingly, the benefit of showing an isolated facial feature was even present when that single feature was the mouth. Overall, the findings showed that focusing on isolated facial features, such as the eyes or mouth, can be a valuable strategy for enhancing identity matching with masked perpetrators.

## Introduction

People generally find it easy to decide whether two simultaneously presented faces are the same person or different people when the faces are familiar. Yet, people struggle to perform the same task when the faces are unfamiliar [1]. The latter task, the verification of an unfamiliar person's identity, is often necessary for border control and criminal investigations. Indeed, errors are frequent (20%), even for trained and experienced passport officers [2] and even under optimal conditions (20–30%), such as using same-day, high-quality photographs of faces, with similar expressions and poses [3–6]. In real-life situations, the conditions are far from optimal with

**Data availability statement:** The datasets generated and analyzed in the current study are available in the Dataverse repository [https://doi.org/10.34894/SLRNAH].

**Funding:** This work was supported by a studentship from the China Scholarships Council (File No. 202108210121) awarded to MZ. The funders had no role in the study design, data collection and analysis, decision to publish, or manuscript preparation.

**Competing interests:** The authors have declared that no competing interests exist.

perpetrators wearing masks to conceal their facial features and avoid recognition. This presents a significant challenge for law enforcement agencies attempting to verify one's identity at border control, building security, and criminal investigations. Until recently, however, this challenge has been relatively underexplored. We aim to address this challenge of identifying masked perpetrators by investigating the effectiveness of matching isolated facial features.

Earlier studies have demonstrated that face masks not only hinder face perception but also disrupt holistic processing, leading to qualitative differences in the processing of masked faces (see, e.g., [7,8]). Holistic processing refers to the integration of individual features into a perceptual whole and characterizes face perception (e.g., [9,10]). Nevertheless, holistic processing does not always come to the advantage of the perceiver. When it is not possible to encode the whole face, such as when a face is masked, allowing a face to be processed holistically may hinder recognition. This is because it is difficult to selectively attend to information in one region of a face without being influenced by information in other regions [11]. For example, one study found that holistic processing is inefficient for localizing featural changes such as detecting changes in eyebrows [12]. Another study has suggested that encoding parts of a face (i.e., the area around the eyes) improves recognition of those parts when presented in isolation rather than in the context of a full face [13]. In other words, at times the whole face seems to distract participants from focusing on the encoded facial feature.

The limitations of holistic processing have led some researchers to propose the use of feature-based processing strategies as a potential solution [14,15]. Feature-based processing involves piecemeal processing of each feature in isolation, relying more on one-to-one comparison of local properties [16,17]. Proponents of feature-based processing suggest that face matching can be improved by prompting observers to use diagnostic features for recognition. For example, encouraging participants to focus on specific facial features (such as the eyebrows) increased face-matching performance from 82% to 90% [14]. Likewise, asking participants to rate the similarity of 11 facial features before deciding whether a pair of faces depicted the same person or different people improved identity discrimination [18,19]. Thus, it seems that encouraging feature-based processing (even in addition to holistic processing) confers specific benefits and can improve face matching performance.

Another way to probe observers to use feature-based processing is by limiting the available facial features. Such approaches have shown some success in the context of eyewitness identifications [20,21]. In a series of four experiments, performance in a masked target-present lineup was superior (70–85%) to performance in a non-masked lineup (54–65%) when participants had encoded a masked face. Similarly, Sagana and Hildebrandt [21] found that participants' ability to identify masked perpetrators increased when participants were presented with photo lineups that only showed the encoded facial features, rather than full faces. These findings are commonly interpreted in the context of transfer appropriateness, because of the memory aspect of the identification task. That is, memory performance is improved when

the encoding context matches the retrieval context [22]. Yet, these findings also provide compelling support for using feature-based processing to enhance recognition accuracy for masked faces [14,19].

Previous studies have explored issues related to masked face recognition [7,8,21,23,24], however, most of them have focused on surgical masks, which obscure only a small portion of the face, leaving a large part of the face still visible. In contrast, full face coverings, such as those used by perpetrators, often leave only minimal portions of the face visible, such as the eyes. These more substantial masks present a greater challenge for recognition, as they conceal a larger number of diagnostic features [25,26]. In such cases, relying on isolated facial features rather than holistic processing may enhance matching performance, particularly when only limited portions of the face are available [14,19].

The effectiveness of feature-based processing may depend on the specific task and stimuli used. Face matching involves a one-to-one comparison of two images and does not involve the retrieval of faces from memory. Indeed, recent studies in face matching have presented contrasting findings. The degree of correspondence between the two faces within a pair, termed contextual congruency, can improve matching performance (e.g., [27,28]). However, its impact is contingent upon the specific stimuli involved [7]. This highlights the need for further exploration and understanding of the role of congruency in face matching.

Not all facial features are equally important when identifying individuals (e.g., [8,29,30]). It seems that the upper half of the face, particularly the eye region, plays a critical role in accurate face recognition [31–33], whereas lower face features such as the nose, mouth, and chin are less important [34–36]. The varying importance of the upper and lower regions of the face is reflected in studies showing that covering the eyes leads to more recognition errors compared to covering the mouth ([34]; for similar findings, see [37]). On the other hand, the mouth as a feature is central to a face because it provides communication cues (e.g., emotional expressions, [38,39]). Taken together, these findings highlight the need for further investigation and understanding of the role of feature-based processing in face matching.

The aims of the current line of research are to investigate the advantage of displaying isolated facial features for matching masked faces and to test whether this advantage holds for masked targets that reveal either their eyes or their mouth under extreme occlusion. In two experiments, participants performed a face-matching task. Participants viewed pairs of images featuring unfamiliar faces and determined whether the images depicted the same person or two different people. The images were divided into two types of pairs: congruent and incongruent. In congruent pairs, both the target and the probe images were presented in a similar way, showing the same facial features, such as the full-face (full-full condition) or a single feature (masked-partial condition). Incongruent pairs were presented in dissimilar ways, such as full-face to masked face or single feature (partial) to full-face combinations. In Experiment 1, we displayed the eyes in the partial and masked conditions, in Experiment 2, the presented feature was the mouth. Based on both empirical studies and theoretical foundations showing the advantages of feature-based processing [14,18,30,40], we expected higher matching performance in congruent (full-face – full-face; masked face – partial face) than incongruent pairs (full-face – partial face; masked face – full-face; Hypothesis 1). Table 1 presents a summary of all hypotheses.

**Table 1. Overview of the hypotheses for Experiment 1 and Experiment 2.**

| | |
|---|---|
| **Hypothesis 1** | Higher matching performance in congruent than incongruent pairs |
| **Hypothesis 2** | Higher confidence for congruent than incongruent trials |
| **Hypothesis 3** | Higher confidence for accurate than inaccurate trials |
| **Hypothesis 4** | Faster decisions for congruent than incongruent pairs |
| **Hypothesis 5** | Faster decisions for accurate than inaccurate trials |

## The Confidence-Accuracy Relationship for Matching Faces

In eyewitness identification and face recognition research, the level of confidence expressed after a positive identification or recognition decision can serve as an indicator of accuracy [41–43]. Despite the considerable attention to the confidence-accuracy relationship in the eyewitness and face recognition literature, the confidence-accuracy relationship in face matching remains largely unexplored. The few existing studies suggest that in face matching, confidence is a good indicator of accuracy for both positive ("same person") and negative ("different people") decisions, when the proportion of matched and mismatched pairs in the task is equal [44,45]. However, to the best of our knowledge, no study to date has examined the confidence-accuracy relationship for face matching when the targets were masked.

For masked targets, the existing body of research primarily consists of eyewitness studies [20,23]. These studies have consistently shown that post-identification confidence increases when conditions at recall matched those at initial perception (i.e., contextual congruency), compared to when conditions at recall did not match those at initial perception (i.e., contextual incongruency). Furthermore, confidence ratings from masked and unmasked lineups yielded similar diagnostic values [23]. In other words, although the masked lineup increased confidence in identification decisions, it had no effect on the diagnostic value of confidence.

Given the significant gaps in the literature on the confidence-accuracy relationship for masked targets, particularly in the face-matching domain, we asked participants to rate their confidence after each face-matching decision. We expected participants to be more confident in decisions made in congruent than incongruent pairs (Hypothesis 2). We also expected to find the typical confidence-accuracy relationship, with higher confidence for accurate than inaccurate trials (Hypothesis 3).

## Decision Time-Accuracy Relationship for Matching Faces

Much like the confidence-accuracy relationship, research on eyewitness identification has shown a strong relationship between decision time and identification accuracy [46–48]. For positive identification decisions, shorter decision times tend to be associated with higher levels of accuracy. Evidence in face recognition is limited but the existing studies indicate a stronger negative relationship between response latency and accuracy for positive, as opposed to negative recognition decisions [42]. The decision time-accuracy relationship appears to hold also for masked targets when the retrieval conditions matched the encoding conditions [20]. Specifically, for target identifications, contextual congruency has been associated with faster decision times than contextual incongruency. Nevertheless, this effect did not extend to foil identifications and lineup rejections [20].

To date, no study has investigated the relationship between decision time and accuracy in face matching. To address this research gap, we measured participants' decision time after each matching decision. We expected participants to be faster for decisions made in congruent than incongruent pairs (Hypothesis 4). We also expect faster decision time for accurate than inaccurate trials (Hypothesis 5).

## Method

The recruitment for Experiment 1 began on 17th March 2022 and ended on 30th May 2022, while the recruitment for Experiment 2 began on 15th November 2022 and ended on 30th January 2023. Both experiments received ethical approval by the Ethics Review Committee of the Faculty of Psychology and Neuroscience of Maastricht University (approval codes 245_155_11_2021 and 245_155_11_2021_S2). All participants consented to participation after receiving an approved information sheet. Both experiments were pre-registered prior to data collection on the Open Science Framework (Experiment 1: https://osf.io/nzd3k; Experiment 2: https://osf.io/u4h7t). The datasets generated and analyzed in the current study are available in the Dataverse repository [https://doi.org/10.34894/SLRNAH].

## Experiment 1

### Participants

We used MorePower [49] to determine the required sample size for repeated measures ANOVA with a within–between interaction. The analysis indicated that 128 participants were required to detect a medium effect size ($\eta^2_p$) of .06 using the standard $p = .05$ and power = .80. In total, 170 participants took part in our experiment. Of these, we excluded 29 participants with incomplete data, 19 non-Caucasian participants to minimize the cross-race effect [50], one who reported a technical problem, and one who reported having completed the Glasgow Face Matching Test task previously. The final sample size consisted of 120 participants: 22 males and 98 females with a mean age of $M = 21.03$ years ($SD = 2.19$; age range: 18–32). All participants were recruited via the research participation system of the university and took part voluntarily in return for course credit.

### Design

We used a mixed factorial design. The independent variables were a) contextual congruency (congruent vs. incongruent pairs; within-subjects factor), b) match condition (match vs. mismatch; within-subjects factor), c) participant gender (female vs. male; between-subjects factor) and d) target gender (female vs. male; within-subjects factor). Participants completed all four face matching conditions: full-full (two full faces); masked-partial (one face wore a mask and the other showed the facial part); full-partial (one face showed the full face and the other showed the facial part); masked-full (one face with a mask and the other without a mask).

We measured participants' matching performance as overall matching accuracy across trials and using signal detection theory indices [51,52]. We used the proportion of hits (correctly responding "the same" on a match trial) and false alarms (incorrectly responding "the same" on a mismatch trial) to calculate sensitivity ($d' = z(\text{hit}) - z(\text{false alarm})$) and response bias (criterion $c = -[z(\text{hit}) + z(\text{false alarm})]/2$)). In cases where participants' responses were either 100% accurate or inaccurate, we applied a customary $1/(2N)$ correction rule to the calculation of the $d'$ and $c$ ( [53]; for discussion see [54]; [55]). Therefore, perfect accurate scores (hit = 1) were converted using the formula $1 - 1/(2N)$, and perfect inaccurate scores (false alarm = 0) were converted using the formula $1/(2N)$. Sensitivity measures the ability to distinguish matches from mismatches (i.e., higher sensitivity means greater face-matching performance). Response bias (criterion $c$) is an indicator of a bias towards declaring a match or a mismatch. A positive c value indicates a tendency to declare a "mismatch" pair (conservative bias), whereas a negative value indicates a tendency to declare a "match" pair (liberal bias). Confidence was measured using a sliding scale from 0 (= *not confident at all*) to 100 (= *absolutely positive*). Decision time was automatically measured in seconds.

### Materials

**Stimuli.** We used 80 pairs of female and male Caucasian faces as stimuli. Half of these were identity matches (40 matches), and the other half were identity mismatches (40 mismatches). All images were gathered from the Glasgow Face Matching Test (GFMT; [56]). All faces were presented in greyscale, in a frontal pose, with a neutral expression and a plain background. The maximum size for a face on the screen was 300 × 500 pixels. For the masked conditions, we used photo editing software (Photoshop) to superimpose masks over the original face stimuli.

### Procedure

Participants were tested online, and the entire session took approximately 20 minutes to complete. After giving consent, participants completed the face-matching task. The task was divided into four blocks, each representing a different experimental condition: full-face -full-face; mask face-partial face; full-face -partial face; mask face-full-face. Blocks were presented in random order, followed by a 20-sec break. Each block had 20 trials, 10 matches and 10 mismatches. In each

trial, participants saw a pair of faces with the question "Are they the same person?" appearing beneath the pair. Partici- pants determined whether the face pairs were identity matches (i.e., the same person) or mismatches (i.e., two different people) by pressing one of the two buttons ("same" or "different"). This display remained onscreen until a response was made. Decision time was recorded automatically. After each trial, participants provided a confidence judgment for their responses. Participants also answered two attention check questions. The first attention check appeared following the 16th trial in the first block, and the second question appeared following the third trial in the third block. At the end of the experiment, participants were thanked and debriefed.

## Analyses

**Contextual Congruency.** To test the effects of contextual congruency on matching performance (Hypothesis 1), we pre-registered and used repeated measures ANOVA with contextual congruency as factor. To examine potential differ- ences between the four conditions (full-full, masked-partial, full-partial and masked-full) and not just as a function of con- gruency (congruent vs. incongruent), we used paired samples *t*-tests. The paired samples *t*-tests were not preregistered. However, these analyses allow us to identify where one or more of the conditions primarily drive observed effects, which is not possible when relying solely on aggregated conditions. Therefore, they contribute to our theoretical understanding.

**Post-Decision Confidence and Decision Times.** To explore the impact of masked face matching on confidence (Hypotheses 2–3) and decision time (Hypotheses 4–5), we pre-registered and conducted two separate repeated measures ANOVAs. We included contextual congruency, accuracy, and their interaction term as factors in the analysis. Due to skew- ness, decision time was normalized using a log-10 transformation. Tables 2 and 3 present the back-transformed values.

In addition to the ANOVAs, to deepen our understanding of the relationship between confidence and accuracy, we plotted calibration curves [43] and conducted ROC analysis. We did not preregister the calibration analysis, but it falls into the same logic as the ROC analysis, which was pre-registered. For the calibration analysis, we plotted the proportion of correct matching decisions across six bins of confidence (up to 50%, 51%−60%, 61%−70%, 71%−80%, 81%−90%,

**Table 2. Face matching performance in the four experimental conditions in Experiment 1.**

| | Congruent | | | | | | Incongruent | | | | | |
|---|---|---|---|---|---|---|---|---|---|---|---|---|
| | Full-full[a] | | | Masked-partial[b] | | | Full-partial[c] | | | Masked-full[d] | | |
| | M | SD | 95% CI | M | SD | 95% CI | M | SD | 95% CI | M | SD | 95% CI |
| **Accuracy** | 0.91 | 0.08 | [0.90, 0.92] | 0.82[a***] | 0.11 | [0.80, 0.84] | 0.80[ab***] | 0.11 | [0.78, 0.82] | 0.72[abc***] | 0.13 | [0.70, 0.74] |
| **Hit Rate** | 0.93 | 0.11 | [0.91, 0.95] | 0.80[a***] | 0.21 | [0.77, 0.83] | 0.73[ab***] | 0.21 | [0.70, 0.76] | 0.62[abc***] | 0.24 | [0.58, 0.66] |
| **FA Rate** | 0.12 | 0.14 | [0.10, 0.14] | 0.16[a*] | 0.15 | [0.14, 0.18] | 0.15[a*] | 0.13 | [0.13, 0.17] | 0.18[a*** c**] | 0.14 | [0.16, 0.20] |
| ***d'*** | 2.93 | 0.75 | [2.82, 3.04] | 2.19[a***] | 0.84 | [2.06, 2.32] | 1.95[a***b**] | 0.84 | [1.82, 2.08] | 1.39[abc***] | 0.98 | [1.24, 1.54] |
| ***c*** | −0.11 | 0.45 | [-0.18, -0.04] | 0.06[a***] | 0.57 | [-0.03, 0.15] | 0.20[a***b**] | 0.53 | [0.12, 0.28] | 0.30[ab***] | 0.52 | [0.22, 0.38] |
| **DT** | 4.68 | 0.22 | [4.37, 5.01] | 4.37[a*] | 0.21 | [4.07, 4.68] | 4.79[b**] | 0.22 | [4.47, 5.13] | 5.01[a*b***] | 0.22 | [4.68, 5.37] |
| **Confidence** | 75.54 | 12.69 | [73.63, 77.45] | 70.27[a***] | 14.21 | [68.13, 72.41] | 66.86[ab***] | 14.23 | [64.72, 69.00] | 67.03[ab***] | 14.78 | [64.80, 69.27] |

*Note.* FA rate = false alarm rate, *d'* = sensitivity, *c* = response bias, DT = decision time (back transformed from log-10). The table presents the mean hit and false alarm values across all participants and trials without any corrections. However, the mean sensitivity and response bias values are calculated using the 1/(2*N*) correction.

The superscript letters indicate significant differences from:

[a] = full-full,

[b] = masked-partial,

[c] = full partial,

[d] = masked-full,

at *p < .05, **p < .01, ***p < .001.

**Table 3. Face matching performance in the four experimental conditions in Experiment 2.**

| | Congruent | | | | | | Incongruent | | | | | |
|---|---|---|---|---|---|---|---|---|---|---|---|---|
| | Full-full[a] | | | Masked-partial[b] | | | Full-partial[c] | | | Masked-full[d] | | |
| | M | SD | 95% CI | M | SD | 95% CI | M | SD | 95% CI | M | SD | 95% CI |
| Accuracy | 0.91 | 0.08 | [0.90, 0.92] | 0.81[a***] | 0.09 | [0.80, 0.82] | 0.78[a***b*] | 0.10 | [0.77, 0.79] | 0.64[abc***] | 0.09 | [0.63, 0.65] |
| Hit Rate | 0.88 | 0.13 | [0.86, 0.90] | 0.71[a***] | 0.18 | [0.68, 0.74] | 0.68[a***] | 0.20 | [0.65, 0.71] | 0.48[abc***] | 0.15 | [0.46, 0.50] |
| FA Rate | 0.07 | 0.10 | [0.06, 0.08] | 0.10[a**] | 0.12 | [0.08, 0.12] | 0.11[a***] | 0.13 | [0.09, 0.13] | 0.19[abc***] | 0.13 | [0.17, 0.21] |
| $d'$ | 2.90 | 0.78 | [2.79, 3.01] | 2.03[a***] | 0.66 | [1.93, 2.13] | 1.90[a***] | 0.73 | [1.79, 2.01] | 0.92[abc***] | 0.59 | [0.84, 1.01] |
| c | 0.13 | 0.39 | [0.07, 0.19] | 0.36[a***] | 0.46 | [0.29, 0.43] | 0.39[a***] | 0.51 | [0.32, 0.46] | 0.54[abc***] | 0.40 | [0.48, 0.60] |
| DT | 3.98 | 0.21 | [3.72, 4.27] | 3.55[a***] | 0.19 | [3.31, 3.80] | 3.72[ab*] | 0.19 | [3.47, 3.98] | 3.80[b***] | 0.19 | [3.55, 4.07] |
| Confidence | 79.02 | 11.33 | [77.38, 80.66] | 69.97[a***] | 13.02 | [68.09, 71.85] | 62.23[ab***] | 13.06 | [60.34, 64.12] | 66.98[ab**] | 12.91 | [65.11, 68.85] |

*Note.* FA rate = false alarm rate, $d'$ = sensitivity, c = response bias, DT = decision time (back transformed from log-10). The table presents the mean hit and false alarm values across all participants and trials without any corrections. However, the mean sensitivity and response bias values are calculated using the $1/(2N)$ correction.

The superscript letters indicate significant differences from:

[a] = full-full,

[b] = masked-partial,

[c] = full partial,

[d] = masked-full,

at [*]$p < .05$, [**]$p < .01$, [***]$p < .001$.

91%−100%; for a similar approach, see [44]). For the ROC analysis, we used the ROC toolbox [57] to fit the unequal-variance signal detection (UVSD) model [58]. This model examines the response frequencies for correct and incorrect answers separately for matched and mismatched conditions. To determine response bias and discriminability at different confidence levels, the confidence ratings were divided into 20 bins in descending order (10 to −10), ranging from "Sure Same Person" to "Sure Different People". Correct responses were reported more frequently than incorrect responses. The Area Under the Curve (AUC) was used to compare the accuracy of different conditions, with higher AUC values indicating better performance. We used the Delong test [59] to test for statistically significant differences between AUCs.

**Gender Differences.** As several previous studies have consistently found gender differences in face recognition [60–65], we explored whether these findings transfer to masked perpetrators. However, own-gender bias is highly asymmetrical, meaning that females are more accurate at recognizing female than male faces, but males are accurate at the recognition of male and female faces [63–65]. When it comes to face-matching, Megreya et al. [60] found that women were more accurate when matching female faces than male faces, but this was not the case for men. Eye-tracking studies also provided evidence that women gaze longer and gave more fixations to the eyes when viewing own-gender faces [61,64]. Therefore, we have also preregistered exploratory ANOVAs on the effect of gender on matching performance. We conducted a mixed ANOVA with accuracy as the outcome variable and participant gender, target gender, and congruency as factors.

## Results

### Face Matching Accuracy

Table 2 provides descriptive statistics for all dependent measures. As expected (Hypothesis 1), matching accuracy differed significantly as a function of contextual congruency, $F(1, 119) = 197.03$, $p < .001$, $\eta^2_p = .62$. Matching decisions for congruent pairs ($M = .86$, $SE = .01$) were more accurate than matching decisions for incongruent pairs ($M = .76$, $SE = .01$). As expected, participants performed significantly better in the full-full condition than any other condition, all $ts(119) \geq 8.30$, $ps < .001$, $ds \geq 0.94$. Importantly, the masked-partial condition outperformed both the masked-full, $t(119) = 9.34$, $p < .001$, $d = 0.83$, and the full-partial condition, $t(119) = 2.73$, $p = .007$, $d = 0.18$. Consistent with our hypothesis, these results show

that participants benefited more from matching facial features rather than a full-face when the target face was masked. Finally, participants performed better in the full-partial condition than in the masked-full condition, $t(119) = 7.03$, $p < .001$, $d = 0.66$, despite both conditions portraying an identical number of facial features.

### Sensitivity and Response Bias

Consistent with matching performance, participants showed higher sensitivity for congruent ($M = 2.56$, $SE = 0.06$) than incongruent pairs ($M = 1.67$, $SE = 0.07$), $F(1, 119) = 228.04$, $p < .001$, $\eta^2_p = .66$. Similar to accuracy, sensitivity was significantly higher for full-full condition than for any other condition, all $ts(119) \geq 9.19$, $ps < .001$, $ds \geq 0.93$. Most importantly, participants showed higher sensitivity in the masked-partial condition than in the masked-full condition, $t(119) = 9.28$, $p < .001$, $d = 0.88$, and the full-partial condition, $t(119) = 3.13$, $p = .002$, $d = 0.29$. Finally, sensitivity was higher in the full-partial than the masked-full condition, $t(119) = 6.48$, $p < .001$, $d = 0.61$.

Response bias was also significantly larger for incongruent than congruent pairs, $F(1, 119) = 45.88$, $p < .001$, $\eta^2_p = .28$. For congruent pairs, participants presented a liberal response bias ($M = -0.02$, $SE = 0.04$), whereas for the incongruent pairs, participants presented a conservative bias ($M = 0.25$, $SE = 0.04$). On closer inspection (see Table 2), we see significant differences between the two congruent conditions. Participants gave liberal responses in the full-full condition but were unbiased in the masked-partial condition, $t(119) = 3.37$, $p = .001$, $d = 0.33$. Response bias was more conservative in both the masked-full condition and the full-partial condition than the masked-partial condition, $t(119) \geq 2.69$, $p \leq .008$, $d = 0.25$. However, no difference was found between the masked-partial condition and the full-partial condition, $t(119) = 1.94$, $p = .055$, $d = 0.19$.

### Post-Decision Confidence

Confidence differed significantly as a function of contextual congruency, $F(1, 117) = 54.26$, $p < .001$, $\eta^2_p = .32$. In line with Hypothesis 2, decisions for congruent pairs were made with greater confidence ($M = 72.16$, $SE = 1.24$) than for incongruent pairs ($M = 67.00$, $SE = 1.31$). The main effect of accuracy was not significant, $F(1, 117) = 0.26$, $p = .613$, $\eta^2_p < .01$. Participants rated confidence similarly on accurate ($M = 69.76$, $SE = 1.18$) and inaccurate ($M = 69.41$, $SE = 1.37$) trials. Additionally, the interaction effect between contextual congruency and accuracy was not significant, $F(1, 117) = 1.12$, $p = .293$, $\eta^2_p < .01$.

Moving to differences among the four conditions, decisions in the full-full condition were made with greater confidence than those in any other condition, all $ts(119) \geq 6.48$, $ps < .001$, $ds \geq 0.39$. Likewise, confidence was higher in the masked-partial than in the masked-full condition, $t(119) = 5.28$, $p < .001$, $d = 0.22$, and the full-partial condition, $t(119) = 5.17$, $p < .001$, $d = 0.24$. The difference between the two incongruent pairs (full-partial vs. masked full) was not significant, $t(119) = 0.23$, $p = .815$, $d = 0.01$.

### Confidence-Accuracy Relationship

We created separate calibration curves for positive ("same person") and negative ("different people") decisions. Fig 1 shows that participants' confidence ratings were not well-calibrated either for positive ("same") or for negative ("different") decisions. However, for positive decisions, the calibration curve was higher for congruent than for incongruent pairs, indicating that participants were more accurate in matching congruent (full-full and masked-partial) than incongruent (full-partial and masked-full) face pairs. This result is consistent with our findings on matching performance. Notably, the calibration curve – and thus accuracy – was lowest when participants matched a masked face to a full-face (masked-full condition). The calibration curves for negative decisions show a large overlap, indicating no major differences across conditions.

Next, we performed an ROC analysis. As shown in the left part of Fig 2, the curves for congruent and incongruent trials closely follow the top-left corner, indicating good to excellent discrimination. Additionally, the DeLong test [56] showed

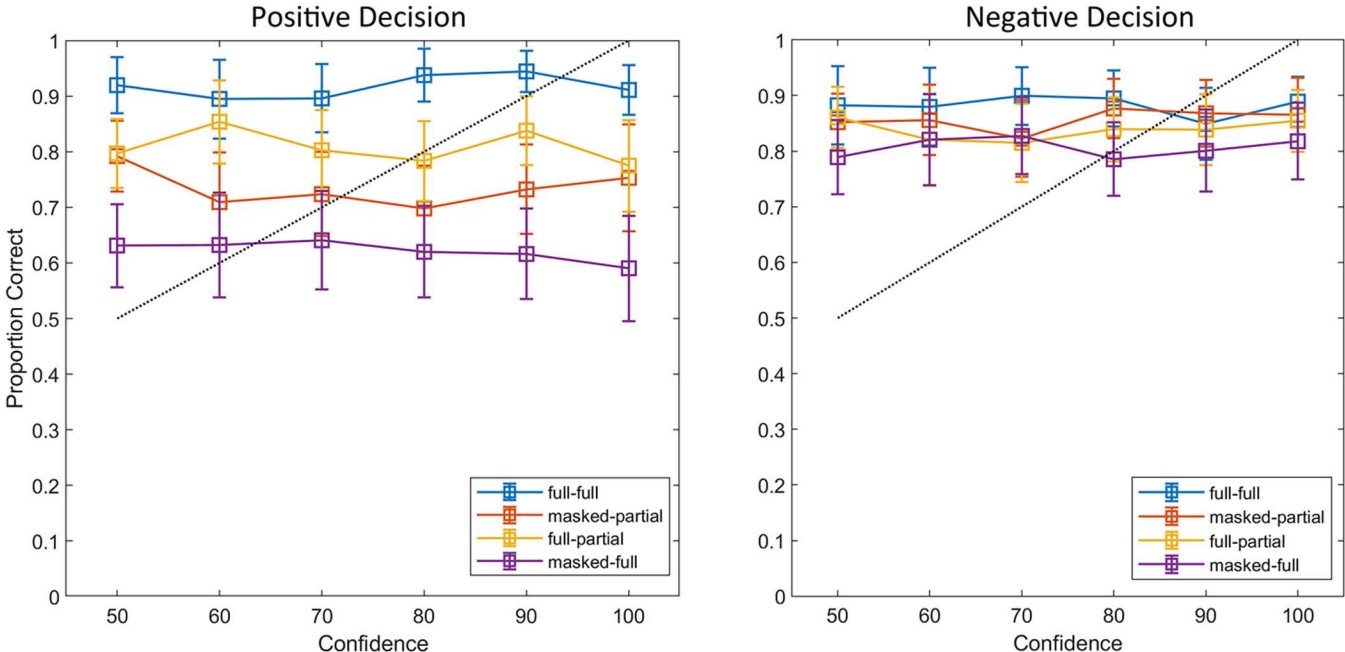

**Fig 1. Calibration curves for positive ("same person", left panel) and negative ("different people", right panel) decisions for each condition in Experiment 1.** The dotted line denotes perfect calibration. Error bars show 95% confidence intervals for the proportions.

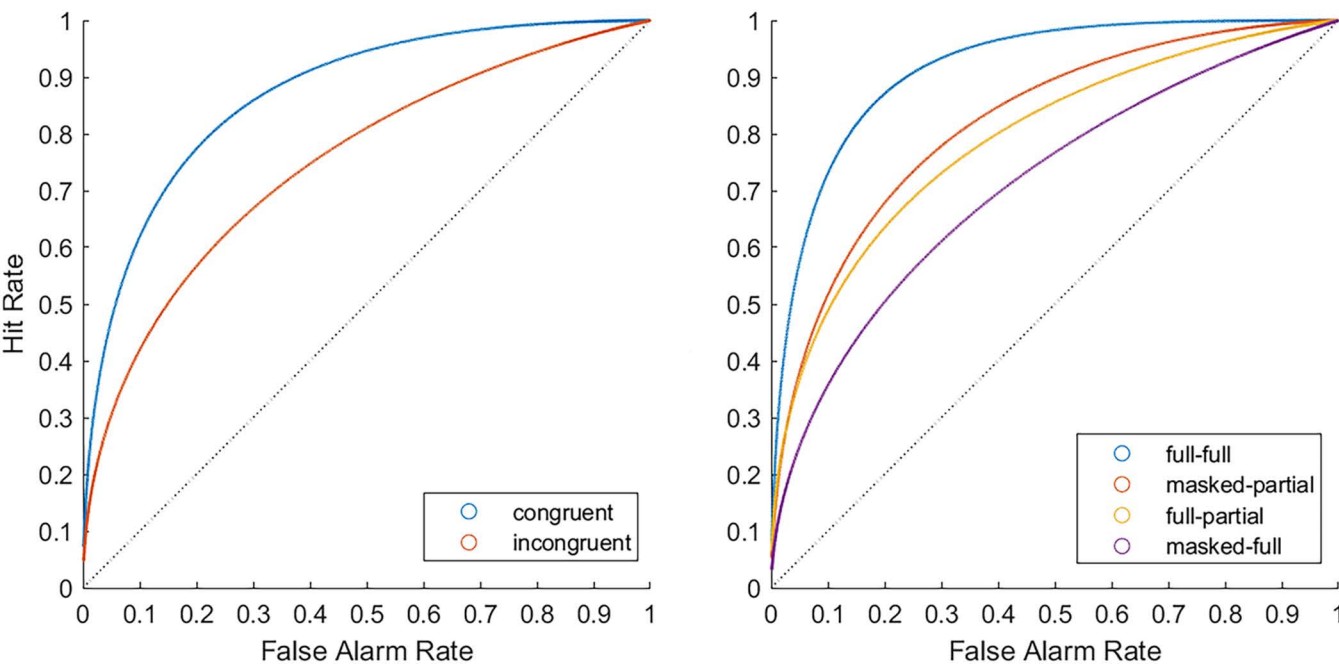

**Fig 2. Confidence-based ROC curves for congruent and incongruent face sets (left) and for each experimental condition (right) for Experiment 1.** The left panel shows the ROC curves for congruent (full-full and masked-partial) pairs versus incongruent (full-partial and masked-full) pairs. The right panel shows the ROC curves separately for each experimental condition. Each point on the plot reflects the hit and false rate at each level of confidence. The diagonal line represents chance performance. The further points were away from the line of chance, the better the discriminability.

better discriminability for congruent (AUC = .87) than incongruent pairs (AUC = .76), $p < .001$. The full-full condition (see right part of Fig 2 ) yielded the best discriminability (AUC = .91) than all other conditions, $ps < .001$. The masked-partial (AUC = .82) condition performed similarly to the full-partial condition = .80), $p = .077$, however, it exhibited higher discriminability compared to the masked-full condition (AUC = .72), $p < .001$. The full-partial condition yielded better discriminability than the masked-full condition, $p < .001$. These findings support our hypotheses 1 and 2 of better matching performance in the congruent pairs than in the incongruent pairs.

### Decision Time

Decision time differed as a function of contextual congruency, $F(1, 117) = 3.93$, $p = .050$, $\eta^2_p = .03$, as predicted by Hypothesis 4. Participants responded faster in congruent ($M = 4.57$, $SE = 0.02$) than incongruent pairs ($M = 4.90$, $SE = 0.02$). Contrary to Hypothesis 5, the main effect of accuracy was non-significant, $F(1, 117) = 1.36$, $p = .247$, $\eta^2_p = .01$. The two-way interaction between the contextual congruency and accuracy was also not significant, $F(1, 117) = 0.71$, $p = .402$, $\eta^2_p < .01$.

Across the four different conditions, participants responded marginally faster in the masked-partial condition than in other conditions, all $ts(119) \geq 2.16$, $ps < .05$, $ds \geq 0.14$. Decision time was faster in the full-full condition than in the masked-full condition, but the effect size is negligible, $t(119) = 2.35$, $p = .021$, $d = 0.14$. No differences emerged between the full-full and the full-partial condition, $t(119) = 0.78$, $p = .436$, $d < 0.01$. Likewise, the difference between the two incongruent conditions (full-partial vs. masked-full) was non-significant, $t(119) = 1.60$, $p = .111$, $d = 0.09$.

### Gender Differences

To explore a potential gender effect, we preregistered and conducted a mixed ANOVA with accuracy as the outcome variable and participant gender, target gender, and congruency as factors. In Experiment 1, the analysis yielded only a significant main effect of facial congruency, $F(1, 118) = 131.74$, $p < .001$, $\eta^2_p = 0.53$. Both male and female participants performed better in congruent ($M = .86$, $SE = .01$) than incongruent sets ($M = .75$, $SE = .01$), irrespective of the target's gender. All other factors and interaction terms were not significant, all $Fs(1, 118) \leq 2.71$, $ps \geq .102$, $\eta^2_p s \leq .02$.

## Experiment 2

Experiment 1 showed better matching performance in congruent than incongruent pairs. The findings highlight the advantages of feature processing and are consistent with transfer appropriateness. In Experiment 2, we investigated whether the mouth could be matched as effectively as the eyes when presented in isolation. The materials, design, and procedure were largely similar to those of Experiment 1, with two differences. First, in Experiment 2, the mask covered the whole face but the mouth instead of the eyes. Second, we replaced the balaclava mask used in Experiment 1 with a black box in Experiment 2.

### Participants

The power analysis indicated that 128 participants were required to detect a medium effect size ($\eta^2_p$) of .06 using the standard $p = .05$ and power = .80. In total, 166 participants took part in the experiment. We excluded 27 participants with incomplete data, 5 non-Caucasian participants to minimize the cross-race effect [50] and 4 participants who reported technical problems. The final sample size consisted of 130 participants: 20 men, 109 women, 1 other gender, with a mean age of $M = 20.14$ years ($SD = 2.03$; age range: 17–27).

### Results

#### Face Matching Accuracy

Table 3 provides descriptive statistics for all dependent measures. The main effect of contextual congruency was significant, $F(1, 129) = 346.02$, $p < .001$, $\eta^2_p = .73$, with better matching accuracy for congruent ($M = .86$, $SE = .01$) than for

incongruent pairs ($M = .71$, $SE = .01$). Once again, participants performed best in the full-full condition, compared to all other conditions, all $ts(129) \geq 10.38$, $ps < .001$, $ds \geq 1.17$. The masked-partial condition yielded higher accuracy than the masked-full condition, $t(129) = 14.73$, $p < .001$, $d = 1.89$, and the full-partial condition, $t(129) = 2.06$, $p = .042$, $d = 0.32$. Finally, participants performed better in the full-partial condition than the masked-full condition, $t(129) = 12.78$, $p < .001$, $d = 1.47$, The results are analogous to those obtained in Experiment 1 and consistent with Hypothesis 1, showing that isolated facial features facilitate matching for masked faces.

### Sensitivity and Response Bias

Sensitivity was higher in congruent ($M = 2.46$, $SE = 0.05$) than in incongruent pairs ($M = 1.41$, $SE = 0.04$), $F(1, 129) = 323.95$, $p < .001$, $\eta^2_p = .72$. As expected, sensitivity was higher in the full-full condition than in any other condition, all $ts(129) \geq 10.99$, $ps < .001$, $ds \geq 1.20$. The masked-partial condition yielded higher sensitivity than the masked-full condition, $t(129) = 14.44$, $p < .001$, $d = 1.77$, but yielded the same sensitivity as the full-partial condition, $t(129) = 1.59$, $p = .113$, $d = 0.19$. Additionally, similar to accuracy, the full-partial condition showed higher sensitivity than the masked-full condition, $t(129) = 12.43$, $p < .001$, $d = 1.48$.

As for response bias, participants displayed more conservative response bias in incongruent pairs ($M = 0.46$, $SE = 0.03$) than congruent pairs ($M = 0.25$, $SE = 0.03$), $F(1, 129) = 49.24$, $p < .001$, $\eta^2_p = .28$. Pairwise comparisons revealed that response bias was smaller in the full-full condition than in any other condition, all $ts(129) \geq 4.58$, $ps < .001$, $ds \geq 0.54$. Likewise, response bias in the masked-partial condition was smaller than the masked-full condition, $t(129) = 4.23$, $p < .001$, $d = 0.42$, but similar to the full-partial condition, $t(129) = 0.61$, $p = .545$, $d = 0.19$. Finally, response bias in the full-partial was smaller than the masked-full condition, $t(129) = 4.23$, $p < .001$, $d = 0.33$.

### Post-Decision Confidence

Confidence differed significantly as a function of contextual congruency, $F(1, 128) = 92.22$, $p < .001$, $\eta^2_p = .42$. Supporting Hypothesis 2 and the findings of Experiment 1, decisions in congruent pairs were made with greater confidence ($M = 74.07$, $SE = 1.01$) than in incongruent pairs ($M = 67.29$, $SE = 1.08$). However, contrary to Hypothesis 3, the main effect of accuracy was not significant, $F(1, 128) = 0.05$, $p = .820$, $\eta^2_p < .01$. Participants were equally confident in accurate ($M = 70.73$, $SE = 0.95$) and inaccurate ($M = 70.63$, $SE = 1.07$) trials. Additionally, the interaction between contextual congruency and accuracy was not significant, $F(1, 128) = 0.98$, $p = .324$, $\eta^2_p < .01$.

Matching decisions in the full-full condition were made with greater confidence than in any other condition, all $ts(129) \geq 8.95$, $ps < .001$, $ds \geq 0.74$. The masked-partial condition yielded significantly higher confidence ratings than the masked-full condition, $t(129) = 3.74$, $p < .001$, $d = 0.23$, and the full-partial condition, $t(129) = 3.50$, $p < .001$, $d = 0.21$. Finally, like Experiment 1, participants' confidence was similar in the full-partial condition and the masked-full condition, $t(129) = 0.31$, $p = .761$, $d = 0.02$.

### Confidence-Accuracy Relationship

The calibration curves for Experiment 2 again revealed that confidence was not a reliable indicator of accuracy, regardless of whether participants made positive or negative decisions (see Fig 3). However, the differences between the four conditions were greater for positive decisions than for negative decisions. Specifically, the calibration curves were higher for full-full and masked-partial conditions compared to the full-partial and masked-full conditions. The calibration curves for negative decisions largely overlap across conditions, indicating small differences between these conditions. Overall, contextual congruency affected the accuracy of mouth matching, but confidence judgments did not always reflect this relationship.

As shown in the left part of Fig 4, the ROC curves for congruent and incongruent pairs were close to the top-left corner, indicating once again good to excellent discrimination. The AUC scores indicated better discriminability for congruent

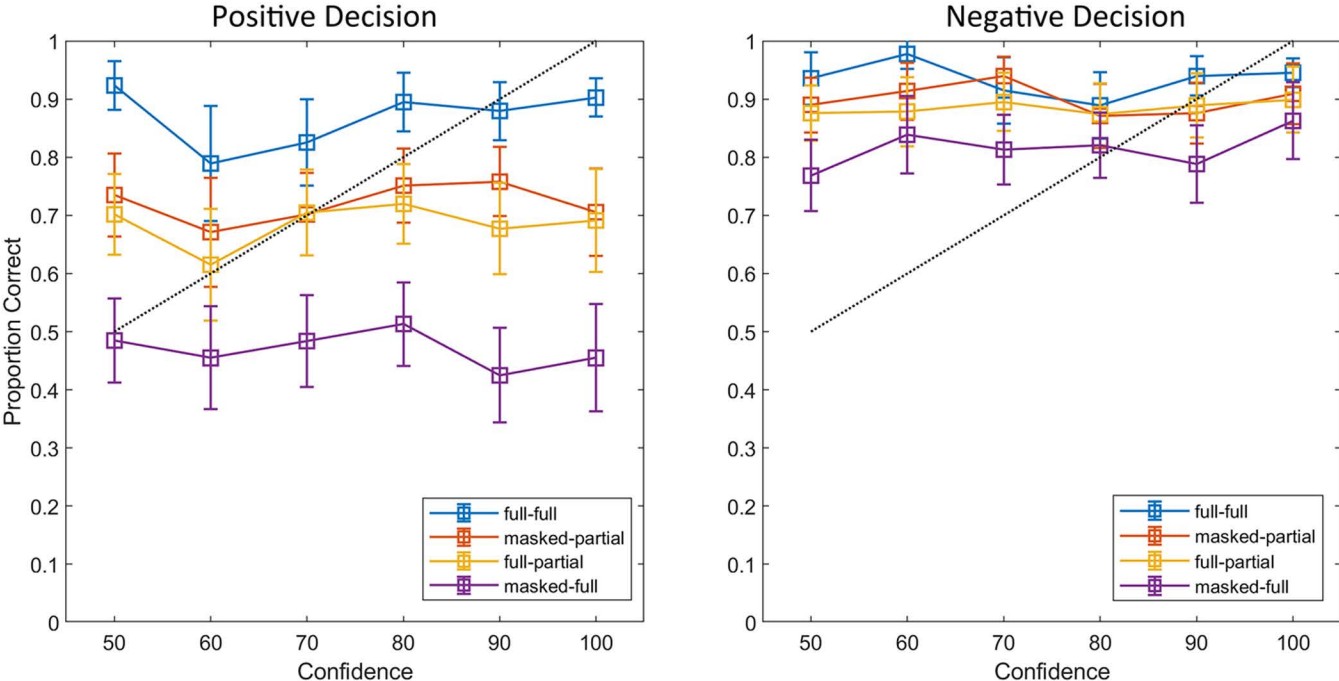

**Fig 3. Calibration curves for positive ("same person", left panel) and negative ("different people", right panel) decisions for each condition in Experiment 2.** The dotted line denotes perfect calibration. Error bars show 95% confidence intervals for the proportions.

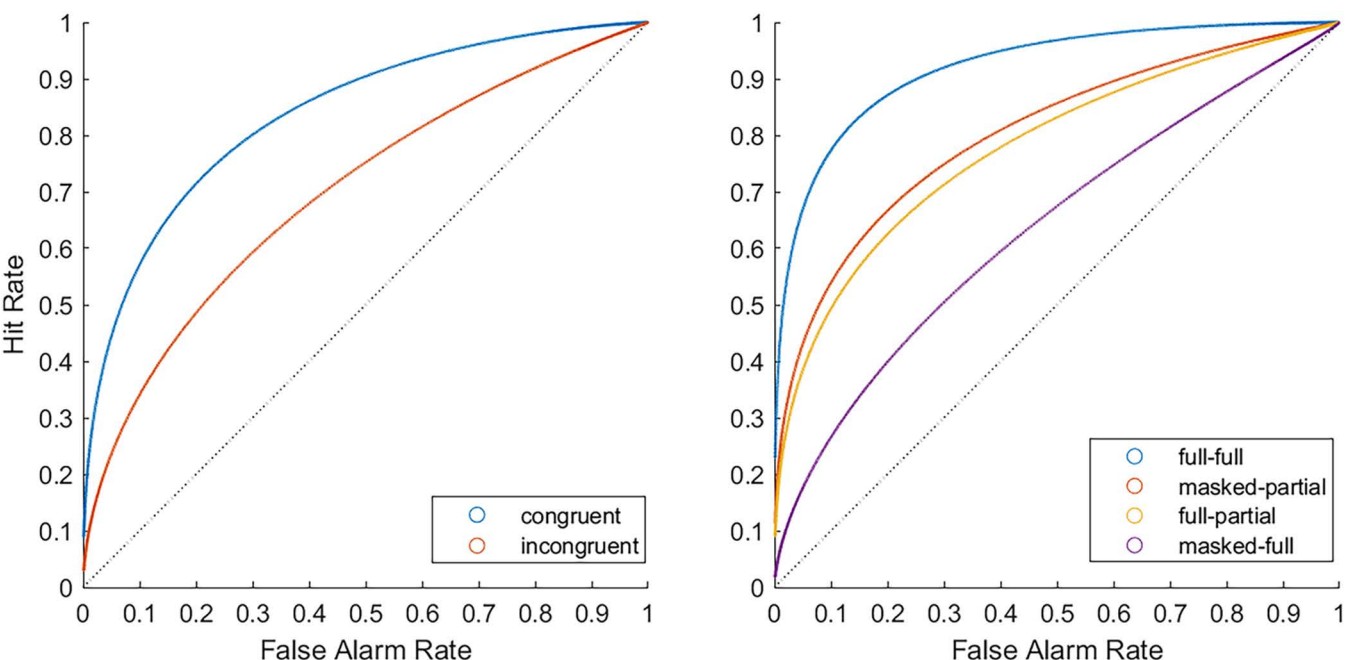

**Fig 4. Confidence-based ROC curves for congruent and incongruent face sets (left) and for each experimental condition (right) for Experiment 2.** The left panel shows the ROC curves for congruent (full-full and masked-partial) pairs versus incongruent (full-partial and masked-full) pairs. The right panel shows the ROC curves separately for each experimental condition. Each point on the plot reflects the hit and false rate at each level of confidence. The diagonal line represents chance performance. The further points were away from the line of chance, the better the discriminability.

(AUC = .84) than incongruent pairs (AUC = .71), $p < .001$. More specifically, the full-full condition yielded the best discriminability (AUC = .91) among all conditions, $ps < .001$ (see right part of Fig 4 ). The masked-partial (AUC = .81) condition yielded better discriminability than the masked-full condition (AUC = .64) and full-partial condition (AUC = .78), $ps \leq .010$. The full-partial condition yielded better discriminability than the masked-full condition, $p < .001$. These findings supported Hypothesis 1 of better matching performance in the congruent than in the incongruent pairs.

### Decision Time

In contrast to Experiment 1 and Hypothesis 4, decision time did not differ as a function of contextual congruency, $F(1, 128) = 1.75$, $p = .188$, $\eta^2_p = .01$. Participants were equally fast in congruent ($M = 3.63$, $SE = 0.02$) and incongruent pairs ($M = 3.72$, $SE = 0.01$). The main effect of accuracy was significant, $F(1, 128) = 4.04$, $p = .047$, $\eta^2_p = .03$, but the effect was in the opposite direction than expected (Hypothesis 5). That is, participants responded faster in inaccurate ($M = 3.63$, $SE = 0.02$) than in accurate ($M = 3.72$, $SE = 0.02$) trials. Finally, the interaction between contextual congruency and accuracy was not significant, $F(1, 128) = 2.82$, $p = .095$, $\eta^2_p = .02$.

Paired samples $t$-tests revealed participants in the masked-partial condition responded the fastest, all $ts(129) \geq 4.36$, $ps < .001$, $ds \geq 0.25$. Likewise, decision time was faster in the full-partial condition than in the full-full condition, $t(129) = 2.35$, $p = .015$, $d = 0.15$. There were no differences between the full-full and the masked-full condition, $t(129) = 1.25$, $p = .213$, $d = 0.08$. There was no difference between the two incongruent pairs (full-partial vs. masked full), $t(129) = 1.43$, $p = .154$, $d = 0.05$.

### Gender Differences

In Experiment 2, we excluded one participant who reported being of other gender. In contrast to Experiment 1, we found a main effect of congruency, $F(1, 127) = 250.35$, $p < .001$, $\eta^2_p = .66$. Participants were more accurate in congruent condition ($M = .86$, $SE = .01$) than incongruent condition ($M = .70$, $SE = .01$). We also found a significant main effect of target gender, $F(1, 127) = 42.00$, $p < .001$, $\eta^2_p = .25$. Participants had better performance when seeing male faces ($M = .81$, $SE = .01$) than female faces ($M = .75$, $SE = .01$). The main effect of participant gender was not significant, $F(1, 127) = 2.79$, $p = .097$, $\eta^2_p = .02$. The three-way interaction between target gender, participant gender, and congruency was not significant, $F(1, 127) = 0.16$, $p = .695$, $\eta^2_p < .01$, and neither were the participant gender × target gender and participant gender × congruency interactions, $Fs(1, 127) \leq 2.38$, $ps \geq .125$, $\eta^2_p s \leq .02$. However, the congruency × target gender interaction was significant, $F(1, 127) = 45.12$, $p < .001$, $\eta^2_p = .26$. Simple main effect analysis on congruent pairs showed that participants were equally accurate at matching male faces ($M = .86$, $SE = .01$) and female faces ($M = .86$, $SE = .01$). For incongruent pairs, participants performed better when presented with male faces ($M = .76$, $SE = .01$) than female faces ($M = .64$, $SE = .01$).

### Discussion

Identity verification of masked perpetrators is a challenging task. Even small changes in facial appearance can affect the ability to match two faces [28]. In two pre-registered experiments, we manipulated contextual congruency to exploit feature-based processing, aiming to enhance performance in matching masked perpetrators. In Experiment 1, we tested the effectiveness of this method for the eyes, and Experiment 2 extended this work to the mouth. The results were consistent with the hypothesis that matching performance improves when the presentation method of both images is congruent rather than incongruent (Hypothesis 1). Most notably, our findings show that presenting isolated features can confer significant benefits in the verification of masked targets for both salient (i.e., eyes) and less salient (i.e., the mouth) facial features.

Matching accuracy and discriminability were markedly higher for congruent than incongruent face pairs in both experiments, generating large effect sizes ($\eta^2_p \geq .62$). Likewise, ROC curves indicated better matching ability for congruent than incongruent face pairs (see Figs 2 and 4). These findings are in line with previous work [7,8,30,66] and transfer

appropriateness [22]. Importantly, unlike previous studies, we have shown that the effect persists even when the number of facial features to be matched is severely restricted. These findings provide strong support for the benefits of featural processing for improving face matching for masked targets [5,66]. Previous research has shown that focusing on specific facial features, such as ears and facial marks, can greatly improve face matching [18], although the advantage of facial features varies [14]. In our study, we found that the advantage of featural processing applied to both the eyes (Experiment 1) and the mouth region (Experiment 2). Therefore, our results suggest that presenting these features separately can be beneficial for both salient and less salient facial features.

Regarding response bias, participants exhibited a more conservative response bias for incongruent pairs than for congruent pairs in both experiments. In Experiment 1, where the mask revealed only the eyes, the response bias varied among the congruent pairs. Specifically, the full-full condition showed a more liberal bias, while the masked-partial condition was neutral. For incongruent pairs, both conditions exhibited a conservative bias, indicating a higher likelihood of declaring a "mismatch". This conservative bias persisted in Experiment 2, where the mask revealed only the mouth, with participants showing a conservative bias for both congruent and incongruent pairs. This conservative bias aligns well with findings showing that face masks lead to "mismatch" responses [7,28,66]. Our findings add to this work by demonstrating that the specific masked facial features may also affect the decision criteria.

We were also among the first to test the role of contextual congruency and matching accuracy on post-decision confidence and decision time. As expected, participants were more confident and responded faster when matching congruent rather than incongruent face pairs, although the effect on decision time occurred only for the eyes (Experiment 1). Together, these findings suggest an ease of processing when matching congruent face pairs. Indeed, when the target and the probe images are presented in similar ways, as showing the same facial features (congruency), one-to-one comparison can not only help improve matching performance [27,28] but also save time [14]. Therefore, these findings support that feature-based processing confers specific advantages.

However, we did not find the typical confidence-accuracy relationship in both experiments [41,43,67,68]. Visual inspection of the calibration curves further supports the lack of a confidence-accuracy relationship for both positive and negative matching decisions despite the large differences. These results provide no support for our hypothesis of a clear relationship between confidence and accuracy for face-matching decisions (Hypothesis 3). The lack of a clear confidence-accuracy relationship in our studies stands in stark contrast to the literature on the confidence-accuracy relationship for face recognition [69], lineups [43,48,67] and face-matching [44]. A possible explanation for the findings could be the perceived difficulty of the task. Participants may have perceived certain conditions as difficult due to the challenging presentation method leading to under-confidence. Indeed, under-confidence may arise from a failure to recognize just how easy a task may be [70]. Consequently, this low level of confidence may have carried over to easier conditions, as the full-full condition where performance is high (see Figs 1 and 3), negatively affecting the confidence-accuracy relationship. The perceived difficulty of the task could also explain the confidence-accuracy relationship in Experiment 2. Indeed, people subjectively perceive mouth matching as a difficult task and therefore do not give self-confidence scores consistent with accuracy [70,71]. Furthermore, participants rate the mouth as less useful for identifying someone than the eyes [15]. Therefore, our participants may have underestimated the diagnostic value of the mouth in face matching and assigned low confidence across their matching decisions.

Another possible explanation for the lack of a confidence-accuracy relationship is individual differences. While largely ignored until recently, individual differences in face recognition appear to modulate the confidence-accuracy relationship. Confidence ratings are more predictive of accuracy in lineup [72] and face recognition tasks [68,73] for people with stronger than weaker face recognition abilities [66,74]. Future research should consider individual differences as a factor in matching performance for masked faces.

We also did not find the typical relationship between decision time and accuracy, where fast decisions are more accurate than slower ones, and the results were inconsistent across experiments. Specifically, decision times did not vary as

a function of accuracy in Experiment 1 and Experiment 2, where participants responded faster on inaccurate trials than accurate trials. The opposite than expected effect in Experiment 2 could be because of reduced scrutiny [75]. Interestingly, in both experiments, participants responded fastest in the masked-partial condition and not in full-full condition. We can speculate that this could be evidence of the disruption of holistic processing in the masked face condition. This phenomenon parallels findings from previous studies on inverted face recognition, where participants exhibited faster reaction times when identifying inverted faces compared to upright faces [10,76].

As for gender differences, although both studies suggest the existence of gender differences, the effect is inconsistent across experiments and the direction of the effect is inconsistent with previous studies [63–65]. Therefore, we advise caution in interpreting these results.

### Limitations and Future Directions

This work is not without limitations. We used laboratory-based face-matching materials that had an idealized masked condition (clear, front face), which closely resembles the standardized photographs found in passports or other official identification documents. However, this differs from masked targets in real-life scenarios. In real life, face matching is even more challenging because of the poor quality of the images [77] and the passage of time. While the increased difficulty in real-life scenarios may lead to a further decline in the relationship between confidence-accuracy and decision time-accuracy, there are no reasons to expect that contextual congruency and feature processing would not provide any benefits. Future studies could investigate this question. We also found that participants performed better in the full-partial condition compared to the masked-full condition, even though both conditions displayed one full-face picture and one facial feature picture. Although the reasons for this finding are unclear, a plausible explanation could be the use of different strategies when matching facial features compared to a masked face. However, this does not detract from the advantages of a congruent partial presentation method over the two incongruent (full-partial, masked-full) conditions. Furthermore, participants rated their confidence lower than their performance which may reflect a failure of metacognitive insight [71,78]. Future studies should focus on the role of the metacognitive processes involved in making decisions based on specific masked facial features. Finally, because human face-matching decisions may be influenced by artificial intelligence (AI) judgments [79] future research may integrate AI and human decision-making, particularly for masked faces, to enhance accuracy and reduce errors in border control systems, where AI alone may struggle with identity resolution [80].

### Conclusion

In two preregistered experiments, we have shown that people can improve face-matching performance by leveraging feature-based processing. Specifically, individuals involved in the recognition of masked perpetrators could benefit from focusing on specific facial features rather than relying on the full face for recognition. Ultimately, the practical relevance of our findings lies in their potential to improve the effectiveness and efficiency of investigations and security measures. This knowledge can contribute to enhancing public safety and security in various contexts, ranging from criminal investigations to border control operations.

### Author contributions

**Conceptualization:** Mengying Zhang, Melanie Sauerland, Anna Sagana.

**Data curation:** Mengying Zhang.

**Formal analysis:** Mengying Zhang.

**Methodology:** Mengying Zhang, Melanie Sauerland, Anna Sagana.

**Project administration:** Anna Sagana.

**Software:** Mengying Zhang.

**Supervision:** Melanie Sauerland, Anna Sagana.

**Validation:** Mengying Zhang, Anna Sagana.

**Visualization:** Mengying Zhang.

**Writing – original draft:** Mengying Zhang, Anna Sagana.

**Writing – review & editing:** Mengying Zhang, Melanie Sauerland, Anna Sagana.

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
