## [Decision Letter · Decision Letter 0]

PONE-D-24-43800Improving Identity Matching for Masked Faces: The Benefit of Isolated Facial FeaturesPLOS ONE

Dear Dr. Zhang,

Thank you for submitting your manuscript to PLOS ONE. After careful consideration, we feel that it has merit but does not fully meet PLOS ONE’s publication criteria as it currently stands. Therefore, we invite you to submit a revised version of the manuscript that addresses the points raised during the review process.

We look forward to receiving your revised manuscript.

Kind regards,

Michal Ptaszynski, PhD

Academic Editor

PLOS ONE

Journal Requirements:

China Scholarships Council, file NO. 202108210121

China Scholarships Council, file NO. 202108210121

China Scholarships Council, file NO. 202108210121

4. Your abstract cannot contain citations. Please only include citations in the body text of the manuscript, and ensure that they remain in ascending numerical order on first mention.

5. We note that Figure 1 and 2 includes an image of a [patient / participant / in the study].

6. We note you have included a table to which you do not refer in the text of your manuscript. Please ensure that you refer to Table 1 in your text; if accepted, production will need this reference to link the reader to the Table.

7. Please include captions for your Supporting Information files at the end of your manuscript, and update any in-text citations to match accordingly. Please see our Supporting Information guidelines for more information: http://journals.plos.org/plosone/s/supporting-information .

Reviewers' comments:

Reviewer's Responses to Questions

**Comments to the Author**

1. Is the manuscript technically sound, and do the data support the conclusions?

Reviewer #1: Partly

Reviewer #2: Yes

2. Has the statistical analysis been performed appropriately and rigorously? 

Reviewer #1: Yes

Reviewer #2: Yes

3. Have the authors made all data underlying the findings in their manuscript fully available?

Reviewer #1: Yes

Reviewer #2: Yes

4. Is the manuscript presented in an intelligible fashion and written in standard English?

Reviewer #1: Yes

Reviewer #2: Yes

5. Review Comments to the Author

Reviewer #1: The manuscript reports Improving Identity Matching for Masked Faces.

However, the subject is interesting field of research, but it is too early to say any facts about the manuscript. There are some extra explanations in some parts and the other part is not clear. In my point of view, there are major concerns as follows;

_ The title should precisely declare which research question is solved, but the title of the manuscript is general. It is recommended to clearly categorize the studied work and present the research aim in a precise title. For example, write the exact problem-solving of the research in the title.

_Abstract is too vague; it contains some sentences but some important aspects of abstract are missing. It should contain problem, challenges, method and evaluation clearly. It is recommended to rewrite the abstract.

__Masked Face identity is introduced in the manuscript but some aspects of facial appearance that is affected on Facial identity matching are missing. For instance, it can be introduced aging or other appearances. Therefore contemplate “ "Toward E-appearance of human face and hair by age, expression and rejuvenation," 2004 International Conference on Cyberworlds, Tokyo, Japan, 2004, pp. 306-311, doi: 10.1109/CW.2004.65. “ then discuss about different appearances.

_Use Natural masked face dataset. If input data has a face mask with the natural looking of human face how it works?

_The comparative study is not complete. Please make a comparative study.

_ The novel contributions are thus not clarified. Please explain where is the novelty of the proposed method.

_How much the proposed study is robust for the face with cosmetics?

Consider “ Toward anthropometric simulation of face rejuvenation and skin cosmetic, https://doi.org/10.1002/cav.38 and then show how the proposed study is robust for real applications?

_ The aim of the proposed method is not adequately explained. It appears that the research question has been previously addressed before.

_ The innovative aspect of the proposed method is not adequately explained.

Reviewer #2: The paper's main aim is to improve the recognition and identification of masked perpetrators. They investigated how to benefit from analyzing parts of the facial features to help in this identification process and how accurate this can lead to enhancing the correctness of the results.

The paper is well presented and organized. Its research domain is of significant importance to law forces and other authorities that benefit from facial recognition or partial/masked recognition. The scope of experiments and their diversity have good coverage of different states of masked faces.

The authors put gender differences in a supplementary document, which is unnecessary. The complete part can be included in the main paper, unless there are some page limitations.

I was hoping to see in the experiments, if possible, a comparison with auto-matching features not based on eyewitnesses and comparing to what extent the accuracy differs from the manual experiments.

6. PLOS authors have the option to publish the peer review history of their article (what does this mean? ). If published, this will include your full peer review and any attached files.

**Do you want your identity to be public for this peer review?** For information about this choice, including consent withdrawal, please see our Privacy Policy .

Reviewer #1: **Yes: ** Azam Bastanfard

Reviewer #2: **Yes: ** Iman M.A. Helal

---

## [Author Response · Author response to Decision Letter 1]

17 Apr 2025

Dear Dr. Ptaszynski,

Thank you for the opportunity to revise our manuscript and the valuable feedback from you and the reviewers.

We have carefully considered all comments and suggestions and have revised the manuscript to meet the requirements of PLoS ONE. Below is a detailed point-by-point response to each of yours and the reviewers' comments. Changes made to the manuscript are highlighted in red for your reference.

Editor’s comments

Thank you for your feedback. We have carefully revised our manuscript to ensure that it fully complies with PLoS ONE's style requirements.

China Scholarships Council, file NO. 202108210121. Please state what role the funders took in the study. If the funders had no role, please state: "The funders had no role in study design, data collection and analysis, decision to publish, or preparation of the manuscript." If this statement is not correct you must amend it as needed. Please include this amended Role of Funder statement in your cover letter; we will change the online submission form on your behalf. Please remove any funding-related text from the manuscript and let us know how you would like to update your Funding Statement. Currently, your Funding Statement reads as follows: China Scholarships Council, file NO. 202108210121.

We note that you have provided funding information that is not currently declared in your Funding Statement. However, funding information should not appear in the Acknowledgments section or other areas of your manuscript. We will only publish funding information present in the Funding Statement section of the online submission form. Please include your amended statements within your cover letter; we will change the online submission form on your behalf.

Thank you for your guidance regarding the funding statement. We have removed all funding-related text from the manuscript and have included the following amended Role of Funder statement in our cover letter: "The funders had no role in study design, data collection and analysis, decision to publish, or preparation of the manuscript."

Given this change, we would appreciate if you could update the Funding Statement in the submission system as follows: Funding Statement: This work was supported by a studentship from the China Scholarships Council (File No. 202108210121).

3. Your abstract cannot contain citations. Please only include citations in the body text of the manuscript, and ensure that they remain in ascending numerical order on first mention.

We reviewed our abstract and confirm that it does not contain any citations (see p. 2).

4. We note that Figure 1 and 2 includes an image of a [patient / participant / in the study].

We have removed the figures related to the facial image dataset from the manuscript. Accordingly, the revised version no longer contains any images of individuals. We believe this revision fully satisfies the journal’s requirements regarding informed consent and the publication of potentially identifiable information.

5. We note you have included a table to which you do not refer in the text of your manuscript. Please ensure that you refer to Table 1 in your text; if accepted, production will need this reference to link the reader to the Table.

Thanks for pointing this out, we have now added a reference to Table 1 in the main text: “Table 1 presents a summary of all hypotheses” (see p. 6).

In response to the second reviewer’s comments, we moved the gender difference analysis from the supplementary to the main text. As a result, the manuscript no longer has a Supporting Information section.

Reviewer 1 comments

1. The title should precisely declare which research question is solved, but the title of the manuscript is general. It is recommended to clearly categorize the studied work and present the research aim in a precise title. For example, write the exact problem-solving of the research in the title.

We appreciate the reviewer’s suggestion regarding the title. Our study aims to examine the effectiveness of presenting isolated features, such as the eyes (Experiment 1) and the mouth (Experiment 2) in enhancing face-matching accuracy for masked targets compared to using the full face. The title clearly states the aim/“problem”: “Improving Identity Matching for Masked Faces” and the method/“problem solving” we used to achieve this “The Benefit of Isolated Facial Features”. However, to enhance precision and address the reviewer’s suggestion, we have revised the title to: "Masked Face Matching Benefits from Isolated Facial Features."

2. Abstract is too vague; it contains some sentences but some important aspects of abstract are missing. It should contain problem, challenges, method and evaluation clearly. It is recommended to rewrite the abstract.

Based on your suggestions, we have revised the abstract to make the introduction of the problem and challenges clearer. Specifically, we have changed the opening to the following: “Verifying the identity of an unfamiliar person is a difficult task, especially when targets wear masks that cover most of their faces. This presents a major challenge for law enforcement in border control, security, and criminal investigations. Therefore, we aim to explore ways to improve face matching performance when a face is heavily masked”. (see p. 2)

Additionally, we have clarified the evaluation method now stating that: “we investigated whether matching isolated facial features, namely the eyes (Experiment 1) and the mouth (Experiment 2), instead of a full face can improve matching accuracy when a target is masked”.

We believe that the methods are presented in a clear way that aligns well with the expectations of psychological journals. Specifically, we describe the stimuli presented and the task of the participants in each condition: “In congruent pairs, participants matched a full-face image to another full-face image or a masked image to an isolated feature. In incongruent pairs, participants matched a full-face image to an image of the eyes or the mouth only or to a masked image.

Finally, we have simplified the sentence structure of the concluding sentence of the abstract to further improve the clarity of the proposed solution to the challenge: “Overall, the two experiments showed that focusing on isolated facial features, such as the eyes or mouth, can be a valuable strategy for enhancing identity matching when dealing with masked perpetrators”.

3. Masked Face identity is introduced in the manuscript but some aspects of facial appearance that is affected on Facial identity matching are missing. For instance, it can be introduced aging or other appearances. Therefore contemplate "Toward E-appearance of human face and hair by age, expression and rejuvenation," 2004 International Conference on Cyberworlds, Tokyo, Japan, 2004, pp. 306-311, doi: 10.1109/CW.2004.65. “ then discuss about different appearances.

We fully acknowledge that factors, such as aging, facial expressions, and other appearance-related changes can indeed affect facial identity matching. However, our work specifically examines the effect of full-face coverings that conceal most facial features, leaving only the eyes and mouth exposed. This setup effectively minimizes the influence of other facial features. Additionally, our work focuses on improving human face-matching performance when masks heavily obscure faces. Although exploring how systems predict appearances behind masks might enhance face matching is an intriguing research direction, it is beyond the scope of our current study.

4. Use Natural masked face dataset. If input data has a face mask with the natural looking of human face how it works?

We agree that exploring identity matching with more natural-looking masked faces is an important direction for future research. We acknowledge this point in the limitations section of the original submission: “We used laboratory-based face-matching materials that had an idealized masked condition (clear, front face), which closely resembles the standardized photographs found in passports or other official identification documents. However, this differs from masked targets in real-life scenarios. In real life, face matching is even more challenging because of the poor quality of the images [77] and the passage of time.” (p. 31)

However, the primary goal of the present study was to test our hypothesis in a controlled laboratory setting, ensuring internal validity and isolating the role of congruency and feature-based processing in masked face recognition. This controlled approach allowed us to establish a clear causal link between these factors and recognition performance without the additional confounds present in real-world data. Establishing this foundation is essential and necessary before extending the findings to more complex, real-world conditions.

While we have since conducted a separate study that examines masked face recognition using real-world images from various angles (see link: https://osf.io/wqvr4/?view_only=f2453783d8444a979cf0dc476c6d2355), that investigation adopts a different methodological framework and addresses distinct research questions that go beyond the scope of the present study. The current paper is focused on establishing the fundamental mechanisms underlying congruency effects in a controlled environment, which is a necessary step before extending these findings to more complex, real-world scenarios.

Regarding naturally masked faces, prior studies such as those of Manley et al. (2019) and Sagana & Hildebrandt (2022) have used stimuli where target faces wore masks. Their findings are comparable to ours, suggesting that seeing the encoded feature in isolation can enhance identification accuracy.

5. The comparative study is not complete. Please make a comparative study.

The comparative study is inherent in the experimental design. Our study systematically compares different approaches through distinct experimental conditions. Specifically, for masked targets, we compared the current standard practice—the benchmark model (masked-full face condition)—with an alternative approach that leverages feature-based processing (masked-partial condition). This comparison revealed a statistically significant improvement in face matching, amounting to a 13% increase in accuracy. Given that our aim was to isolate and assess the impact of feature-based processing in masked face recognition, the comparative aspect of our study is complete and directly addresses the research question.

Furthermore, we extensively compare our findings with prior studies in multiple ways:

1. Comparison with previous research: The literature on the topics we study in this manuscript is vast. Specifically , we refer to relevant studies on masked face recognition, e.g., “Previous studies have indeed explored issues related to masked face recognition [7,8,21,23,24], however, most of them have focused on surgical masks, which obscure only a small portion of the face, leaving a large part of the face still visible. In contrast, full face coverings, such as those used by perpetrators, often leave only minimal portions of the face visible, such as the eyes”. (see p. 5).

On the confidence-accuracy relationship, we refer to: “The few existing studies suggest that in face matching, confidence is a good indicator of accuracy for both positive (“same person”) and negative (“different people”) decisions, when the proportion of matched and mismatched pairs in the task is equal [44,45]. However, to the best of our knowledge, no study to date has examined the confidence-accuracy relationship for face matching when the targets are masked.” (see p. 7)

2. Comparison with previous results: we discussed how our results align with or differ from previous studies (see p. 28 - 31), focusing on various aspects:

a. Congruency effect: “ These findings are in line with previous work [7,8,30,66] and transfer appropriateness [22]. Importantly, unlike previous studies, we have shown that the effect persists even when the number of facial features to be matched is severely restricted. These findings provide strong support for the benefits of featural processing for improving face matching for masked targets [5,66]”.

b. Response bias: “This conservative bias aligns well with findings showing that face masks lead to “mismatch” responses [7,28,66]”.

c. Confidence-accuracy relationship: “The lack of a clear confidence-accuracy relationship in our studies stands in stark contrast to the literature on the confidence-accuracy relationship for face recognition [69], lineups [43,48,67] and face-matching [44]”.

d. Gender differences: “the effect is inconsistent across experiments and the direction of the effect is inconsistent with previous studies [63-65]”.

Finally, we also compare the results of Study 1 and Study 2, which examine different facial features, focusing on various aspects:

a. Congruency effect: “Matching accuracy and discriminability were markedly higher for congruent than incongruent face pairs in both experiments, generating large effect sizes (ηp2 ≥ .62).” (p. 28)

b. Response bias: “This conservative bias persisted in Experiment 2.” (p. 29)

c. Confidence-accuracy relationship: “we did not find the typical confidence-accuracy relationship in both experiments [41,43,67,68].”(p. 30)

d. Decision time - accuracy relationship: “decision times did not vary as a function of accuracy in Experiment 1 and Experiment 2.” (p. 31)

This additional comparison further strengthens the comparative study of the present work, strengthening the theoretical and practical contribution of our work.

6. The novel contributions are thus not clarified. Please explain where is the novelty of the proposed method.

This line of research is aimed at improving face-matching performance in cases involving heavily occluded faces. This is relevant in criminal contexts but largely overlooked in prior studies. While most masked face matching and recognition research focused on pandemic-related surgical masks, we address the more challenging case of full-face coverings that obscure most facial features. Following your suggestion, we have clarified this novel contribution in the main text: “Previous studies have indeed explored issues related to masked face recognition [7,8], however, most of them have focused on surgical masks, which obscure only a small portion of the face, leaving the large part of face still visible. In contrast, full face coverings, such as those used by perpetrators, often leave only minimal portions of the face visible, such as the eyes. These more substantial masks present a greater challenge for recognition

---

## [Decision Letter · Decision Letter 1]

Masked Face Matching Benefits from Isolated Facial Features

PONE-D-24-43800R1

Dear Dr. Zhang,

We’re pleased to inform you that your manuscript has been judged scientifically suitable for publication and will be formally accepted for publication once it meets all outstanding technical requirements.

Kind regards,

Giulio Contemori, Ph.D.

Academic Editor

PLOS ONE

Additional Editor Comments (optional):

The revision process for this manuscript has taken longer than usual due to our inability to re-establish contact with one of the original reviewers of the initial submission. Despite this, I have carefully reviewed the revised manuscript and the authors’ detailed responses to the prior feedback. I am satisfied that the original reviewer comments have been adequately and thoughtfully addressed. The revisions have significantly improved the clarity and rigor of the work, and the manuscript now meets the journal’s standards for publication.

Reviewers' comments:

Reviewer's Responses to Questions

**Comments to the Author**

Reviewer #2: All comments have been addressed

2. Is the manuscript technically sound, and do the data support the conclusions?

Reviewer #2: Yes

3. Has the statistical analysis been performed appropriately and rigorously? 

Reviewer #2: Yes

4. Have the authors made all data underlying the findings in their manuscript fully available?

Reviewer #2: Yes

5. Is the manuscript presented in an intelligible fashion and written in standard English?

Reviewer #2: Yes

6. Review Comments to the Author

Reviewer #2: Thanks for addressing all my comments. I would like to see the continuation of this research in a more challenging environment with more uncontrolled affecting factors.

7. PLOS authors have the option to publish the peer review history of their article (what does this mean? ). If published, this will include your full peer review and any attached files.

**Do you want your identity to be public for this peer review?** For information about this choice, including consent withdrawal, please see our Privacy Policy .

Reviewer #2: **Yes: ** Iman M.A. Helal

---

## [Editor Report · Acceptance letter]

PONE-D-24-43800R1

PLOS ONE

Dear Dr. Zhang,

I'm pleased to inform you that your manuscript has been deemed suitable for publication in PLOS ONE. Congratulations! Your manuscript is now being handed over to our production team.

Kind regards,

on behalf of

Dr. Giulio Contemori

Academic Editor

PLOS ONE